# Fatigue damage reduction in hydropower startups with machine learning

Till Muser[1], Ekaterina Krymova [1] ✉, Alessandro Morabito [2], Martin Seydoux[2] & Elena Vagnoni[2]

As the global shift towards renewable energy accelerates, achieving stability in power systems is crucial. Hydropower accounts for approximately 17% of energy produced worldwide, and with its capacity for active and reactive power regulation, is well-suited to provide necessary ancillary services. However, as demand for these services rises, hydropower systems must adapt to handle rapid dynamic changes and off-design conditions. Fatigue damage in hydraulic machines, driven by fluctuating loads and varying mechanical stresses, is especially prominent during the transient start-up of the machine. In this study, we introduce a data-driven approach to identify transient start-up trajectories that minimize fatigue damage. We optimize the trajectory by leveraging a machine learning model, trained on experimental stress data of reduced-scale model turbines. Numerical and experimental results confirm that our optimized trajectory significantly reduces start-up damage, representing a meaningful advancement in hydropower operations, maintenance, and the safe transition to higher operational flexibility.

As the global community intensifies efforts toward a sustainable future, the shift to renewable energy sources has become a central objective worldwide. The European Union, for instance, has set an ambitious target to derive at least 42.5% of its energy from renewables by 2030, with aspirations to push this to 45% under its Green Deal framework[1]. This is part of a broader global trend towards decarbonization, where long-term scenarios point to even more substantial reductions in greenhouse gas emissions[2,3]. Central to this transformation is the reliance on intermittent renewable energy sources, such as wind and solar power, necessitating the gradual phase-out of traditional fossil fuel-based power plants. This transition requires robust support systems to maintain frequency and voltage stability, emphasizing the need for ancillary services that ensure proper resilience in both power production and consumption. Hydropower is particularly well-suited for this role, given its ability to provide both active and reactive power regulation[4]. The conventional design of hydraulic turbines has traditionally been optimized for high performance during steady-state operations, where the system operates under constant or nearly constant control parameters over time.

However, the demand for ancillary services is projected to rise significantly in the coming decades, altering the typical operating schedules of these machines. To address the increasing need for flexibility, hydroelectric technologies must be adapted to handle the challenges posed by rapid dynamic changes and more frequent start-up sequences. This often entails operating hydraulic machines under off-design conditions, for which the machine components suffer from intense dynamic loads leading to fatigue damage. Historically, limited consideration was given to fatigue damage caused by start-up sequences in the hydraulic turbine design, as their impact over time was considered negligible. After all, prior to the 2000s, Francis turbines were built to handle only a few dozen start-stop cycles per year, unlike the recent decade, where they may experience up to 500 cycles annually due to the significant rise in intermittent energy sources[5]. Fatigue damage refers to the deterioration of a material's structural properties due to the initiation and propagation of cracks under cyclic or fluctuating stresses. In hydraulic machines, such cracks often originate during manufacturing processes, such as casting and welding, or develop over time during prolonged operation under high loading

[1]Swiss Data Science Center, EPFL & ETH Zürich, Andreasstrasse 5, Zurich, Switzerland. [2]Technology Platform for Hydraulic Machines, École Polytechnique Fédérale de Lausanne, Avenue de Cour 33 bis, Lausanne, Switzerland. ✉e-mail: ekrymova@ethz.ch

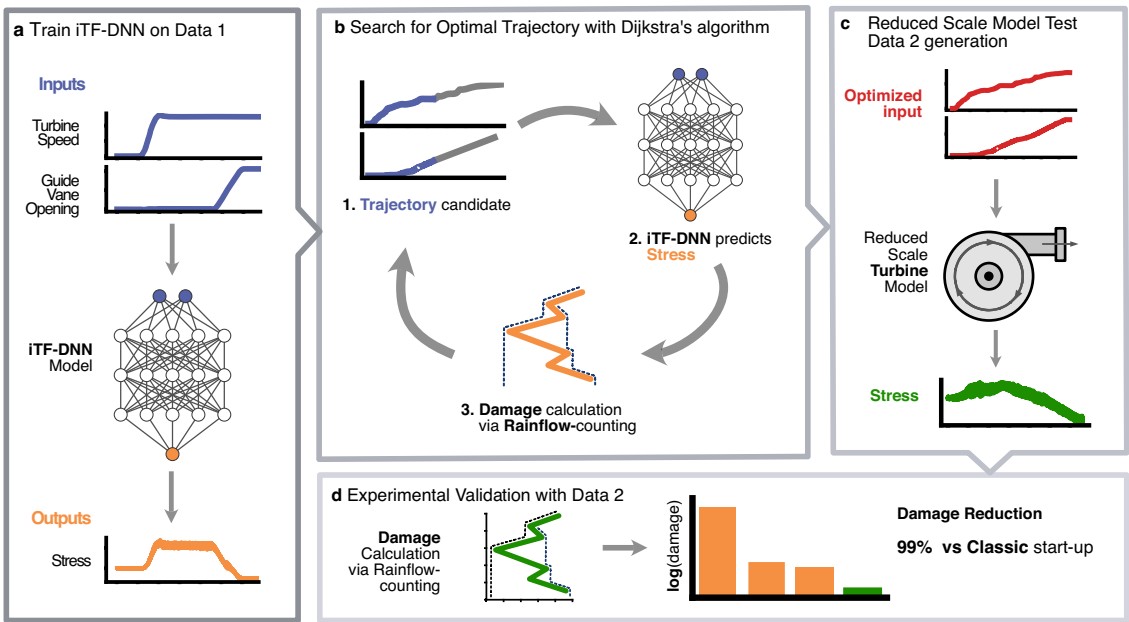

**Fig. 1 | Overview of the workflow steps. a** iTF-DNN is a deep learning model[47] trained on the first dataset of start-up trajectories to predict stress. **b** The optimal start-up trajectory search is based on the damage estimated from the model's stress prediction. **c** The optimized trajectory is tested in the reduced-scale model. **d** Final validation of the optimized trajectory in terms of the damage.

conditions. Many machine components are exposed to fluctuating stresses caused by fluid-structure interactions and vibrations, with runner blades subjected to the most intense loading[6–8]. The fluid-structure interactions and vibrations depend on the operating condition or operating sequence of the hydraulic machine, such as the start-up trajectory, which is defined by its control variables, namely the turbine rotational speed and the guide vanes opening angle (GVO). Therefore, in the context of modernizing the hydropower sector, accurately estimating these stresses and the corresponding induced fatigue is critical for operational optimization and maintenance planning to avoid unexpected failures and outages. Estimating fatigue damage typically involves conducting detailed experiments at the hydropower plant, where mechanical stresses are measured using strain gauges placed on the most stressed parts of the components[9]. This data is then utilized to estimate the reduction in the component's lifespan through fatigue curves. However, this process is cumbersome, as it requires prior numerical simulations to identify the stressed areas and involves the installation of complex equipment, which temporarily halts power production. All these reasons highlight the impracticality of conducting systematically such an experimental campaign on an already commissioned hydropower plant.

Experimental tests on the turbine in *reduced-scale physical models* are usually carried out to verify that the design of the prototype will meet the expected performance. At this stage, specific tests could also estimate local damage and their lifespan. During the experiments, the controlled parameters, such as turbine speed, GVO, and others, are continuously measured, as well as strain stress induced by vibrations. Damage incurred due to fatigue can be further estimated via the Rainflow-counting algorithm[10] followed by Miner's rule[11]. The reduced scale model tests are expensive, and the complexity of the components' geometry makes difficult to create accurate analytical scale models. At the same time, data-driven approaches enable the development of a model for stresses as a function of control parameters, based on the experimental data. Such models can provide further insights into the lifespan of the machine. A study in reduced-scale model Francis turbines has found that transient operations, in particular conventional start-stop sequences, caused severe fatigue, equivalent to many hours or even days at low discharge operation[12,13]

compared to operating at nominal conditions. This is because fatigue damage is largely associated with significant stress alterations[14] on the turbine runner blades, which occur during start-up. Further research demonstrated that modifying the start-up scheme can significantly reduce fatigue damage and extend the life expectancy of Francis turbines[15]. An alternative starting trajectory, made possible by variable speed capable power generators, has been investigated in[16] and has been found to significantly reduce fatigue. Until now, to our knowledge, there has been no study conducted using machine learning methods to optimize start-up trajectories in the hydropower domain.

In this work, we investigate the possibility of the data-driven search for the least damaging transient start-up trajectory. This thorough search of the trajectory space is facilitated by modeling the stress using an input Time-Frequency deep neural network (iTF-DNN) model, trained on data of several start-up trajectory measurements obtained in a dedicated experimental campaign on a reduced scale model complying with IEC standards and fully hydraulically and mechanically homologous to the full scale hydraulic machine at the Hydraulic Machines Platform at the Swiss Federal Institute of Technology Lausanne (EPFL-PTMH)[17]. Relying on the predictions of the iTF-DNN, and the Rainflow-counting/Miner's rule approach to estimate fatigue damage, we employ pathfinding techniques to optimize a start-up control trajectory for predicted incurred fatigue. To evaluate the potential effect of optimization in practice, measurements of induced stresses for the optimized trajectory were recorded on a reduced-scale model at EPFL-PTHM during the second campaign[18]. The results suggest that using the optimized trajectory can significantly reduce the fatigue damage accumulated during the hydraulic machine start-up.

## Results
### Workflow for optimization and validation of start-up trajectory
An overview of the workflow is shown in Fig. 1, which consists of four major steps. First, the stress neural network model is trained on the dataset of the first experimental campaign which contains measurements of the stresses and control inputs for the four known types of start-up trajectories (Fig. 1a). A complete data description of the dataset collection can be found in Methods. Next, the optimal trajectory is identified using Dijkstra's algorithm, relying on fatigue damage

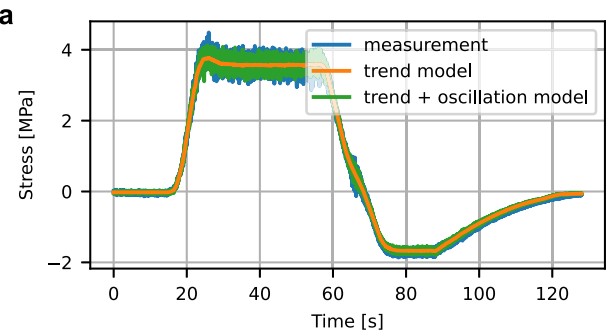

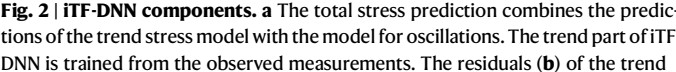

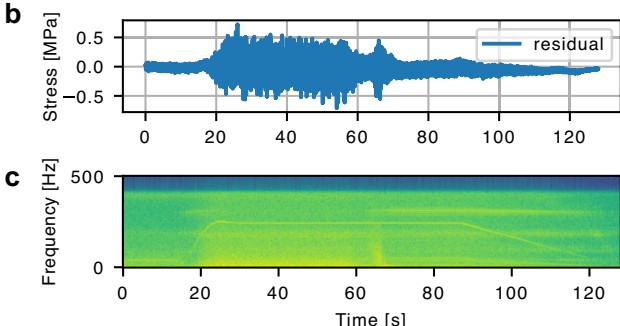

**Fig. 2 | iTF-DNN components. a** The total stress prediction combines the predictions of the trend stress model with the model for oscillations. The trend part of iTF-DNN is trained from the observed measurements. The residuals (**b**) of the trend part of iTF-DNN are transformed with an STFT to the time-frequency domain (**c**) and further used to train the oscillation part of iTF-DNN.

estimates derived from the predicted stress (Fig. 1b). To evaluate the optimized trajectory in the real experiment, it was tested in the reduced-scale model test at EPFL-PTHM (Fig. 1c). Finally, we validate the damage of the obtained stress data of the second experimental campaign and compare the damages of the known trajectories (Fig. 1d).

### Stress model-based damage prediction

We focus on modeling the stress, as it can be directly measured in the turbine. Fatigue damage can be derived via post-processing a stress sequence (see "Methods"). Notably, stress which is measured in the reduced-scale model can be scaled to a full-size hydraulic machine[19]. To model the stress, we introduce the iTF-DNN stress model, which serves as a core component for optimizing stress-induced fatigue damage in our approach.

Consider a decomposition for the observed time series of stresses of the length $T$, $\boldsymbol{y}_t = \boldsymbol{m}_t + \boldsymbol{o}_t + \boldsymbol{\epsilon}_t$, $t = 1, \ldots, T$, where $\boldsymbol{m}_t$ is a slowly varying trend, $\boldsymbol{o}_t$ is an oscillatory component, and $\boldsymbol{\epsilon}_t$ is random noise. The iTF-DNN model consists of two parts: a trend model for $\boldsymbol{m}_t$ trained in the original stress scale, and another model for $\boldsymbol{o}_t$, trained in the time-frequency domains of the stress observations. One of the key assumptions we make is that the slow-varying trend of the stresses depends on the control parameters instantaneously[13]: the trend $\boldsymbol{m}_t$ is deterministic and depends solely on the current input control values GVO, turbine rotational speed, and Head (and not on the control history). Head is set by site conditions, while GVO and turbine rotational speed are the parameters which typical control systems of hydraulic machines, such as Proportional-integral-derivative (PID) controllers, regulate to steer operations. To model the trend part, we trained a neural network that takes control values at time $t$ as input and outputs the stress values by optimizing the mean squared error (MSE). See Fig. 2a for an example of a fitted trend model estimate. The model of the trend produces non-oscillating predictions. The remaining oscillatory component $\boldsymbol{o}_t$ is allowed to depend on the control history in a non-instantaneous way. In particular, we estimate $\boldsymbol{o}_t$ in the time-frequency domain, by modeling amplitudes and phases corresponding to a range of frequencies of the short-time Fourier transform (STFT) of the residual after estimating $\boldsymbol{m}_t$ as a function of the inputs, see Fig. 2b. The estimate of the oscillatory part in iTF-DNN is recovered through the inverse STFT (ISTFT) based on the estimates of the amplitudes and phases of the STFT. The architectures of the models of the trend $\boldsymbol{m}_t$ and of the STFT amplitudes and phases of $\boldsymbol{o}_t$ are based on ResNet[20]). The final iTF-DNN model of stress combines the model of the trend and the model of the oscillation, see Fig. 2a. We describe iTF-DNN more formally in Methods.

To train and validate the model, stress measurements of four types of start-up trajectories were available to us as a result of the XFLEX HYDRO project[16,17]: The standard Classic start-up trajectory, the Linear, BEP and 2Slopes trajectories (see "Methods" for data collection description). Note that the non-classic start-up sequences take advantage of variable speed unit capability and are defined according to a coupled 1D-3D numerical simulations study targeting the optimization of the trajectory by using reduced order models of the hydraulic system[21]. The iTF-DNN is trained on the data of two well-recorded sensors for the Classic, Linear, and 2Slopes startups, and validated on the BEP trajectory. The model demonstrated values of $R^2$ greater than 0.97 on both training and validation datasets, see the details in Methods and Table 3 therein.

As we will further use the stress model to identify an optimal start-up trajectory, we verify that the damage estimates mimic the damage of the measured stress well. Damage comparison in Fig. 3 for the three types of start-ups used for training (Classic, Linear, and 2Slopes trajectories) and testing (BEP trajectory) shows that on average the damage from the predicted stress is proportional to the damage of the measurements. This suggests that the iTF-DNN model predicts the stress from the input controls with enough accuracy to construct a reliable damage proxy, which can then be used for optimization purposes.

### Model-based optimization of the start-up

In optimizing the start-up trajectory, our goal is to minimize damage by adjusting the turbine speed and GVO. The third input, the Head, while used to train the neural network model, is kept constant, as it is assumed to remain static during start-up. For a new candidate $\boldsymbol{u}$, containing the trajectories of controls, we estimate the stress using the iTF-DNN model (denoted as $f_{\mathrm{NN}}$ below) and then evaluate the incurred (scalar) damage as

$$\hat{D}(\boldsymbol{u}) = \mathrm{Damage}\,(f_{\mathrm{NN}}(\boldsymbol{u})). \tag{1}$$

The goal of the optimization is to find the sequence of controls, $\boldsymbol{u}^{\star} \in \mathcal{U}$, that minimizes the predicted fatigue damage

$$\boldsymbol{u}^{\star} = \arg\min_{\boldsymbol{u} \in \mathcal{U}} \hat{D}(\boldsymbol{u}). \tag{2}$$

Here, $\mathcal{U}$ is a space of admissible trajectories, satisfying a number of conditions (see (11)-(13) in Methods) that ensure its feasibility.

For trajectory search, we applied Dijkstra's Algorithm[22], which has a long history of use in mechanical engineering, particularly in trajectory optimization. Recent applications include planning efficient motion paths for robots[23,24], optimizing flight trajectories[25-32], enhancing automated guided vehicle and mobile robots routing[33-36], to cite a few. We reformulate the original problem stated in Eq. (2) as finding the minimal-cost path between two nodes in a weighted graph, making it suitable for the application of Dijkstra's algorithm. To achieve this, we discretize the phase space of GVO and turbine speed, and build a

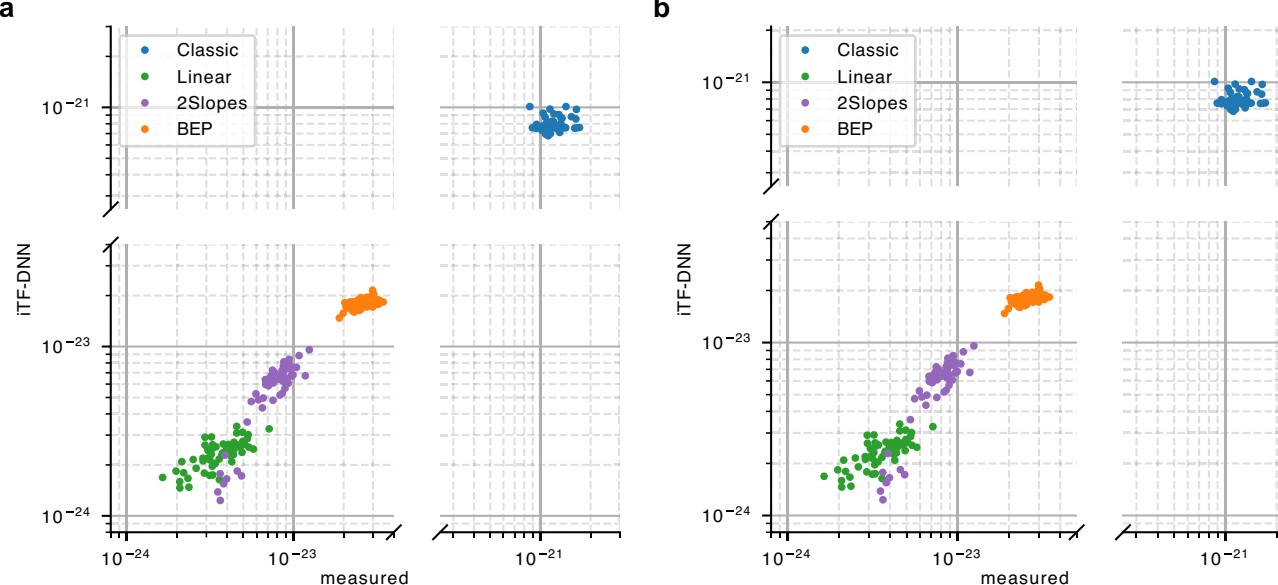

**Fig. 3 | Comparison of damage predictions from iTF-DNN with damage based on measurements.** Damage for sensor 1 (**a**) and sensor 2 (**b**) computed based on the real measurements and predicted from the iTF-DNN. The model was trained on the Classic, 2Slopes, and Linear trajectories, while the BEP trajectory was excluded from the training data.

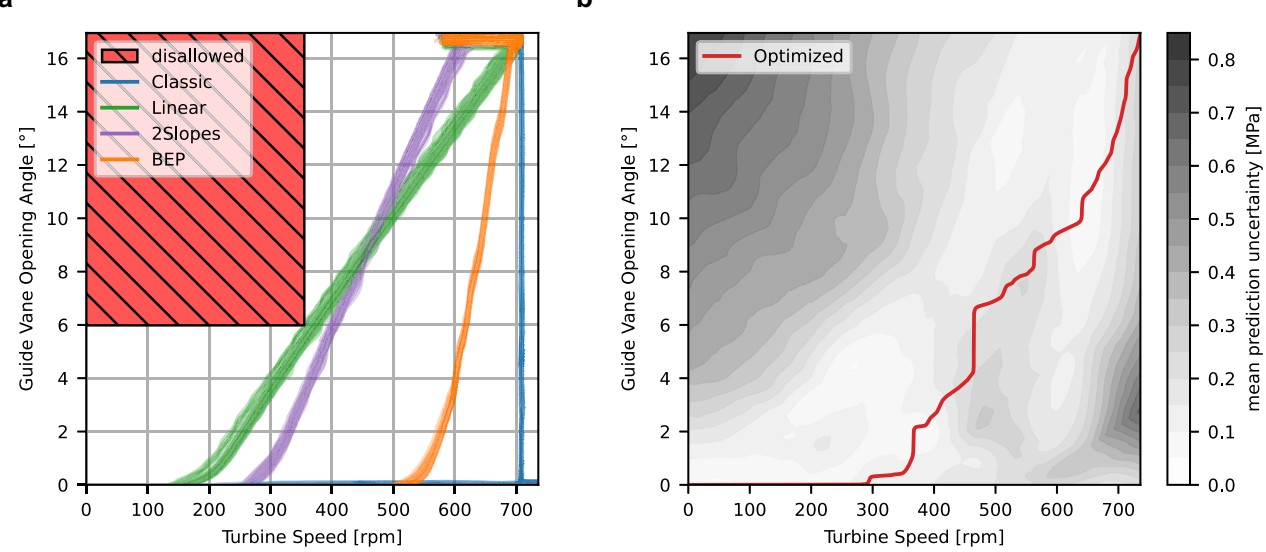

**Fig. 4 | Phase-space plots of the optimization setup and resulting optimized trajectory. a** Data from the first experimental campaign used to train iTF-DNN, along with the disallowed region defining the constraint applied in start-up optimization. **b** The final optimized trajectory, shown alongside the stress model uncertainty.

graph taking into account the conditions for trajectory admissibility. We use the fatigue damage, computed from the stress estimates provided by the iTF-DNN, as a cost function. See "Methods" for the details.

The resulting trajectory is visualized in Fig. 4b, which also includes uncertainty estimates for the trend part of the iTF-DNN model. To obtain uncertainty estimates, we employed a deep ensemble[37–39] consisting of 10 independently trained trend component models of iTF-DNN. Note that, due to the model's instantaneous dependence on the control values, the uncertainty estimates are also independent of the control history. The optimized trajectory passes primarily through regions of low uncertainty, suggesting that the results are likely accurate and that the trajectory is suitable for testing with the reduced-scale model. Note that the results for the optimized trajectory rely on the iTF-DNN, which is trained on limited data. Additionally, uncertainty

estimates are based solely on the trend model; thus, they do not account for potential variability introduced by the oscillation model. A comparison of the predicted damage for trajectories tested during the first experimental campaign with the damage estimates from the optimization result is presented in Table 1. We provide both absolute damage values and damage relative to the Classic baseline trajectory. As expected, the optimized trajectory results in significantly lower estimated damage, with a reduction of over 99% compared to Classic.

### Experimental validation by reduced scale model

To assess the performance of the optimized start-up trajectory experimentally, we performed reduced-scale model tests at the EPFL PTMH platform[18]. Note that the stress sensors that had well-recorded measurements in the first data acquisition used to train iTF-DNN

models and get the optimized trajectory were different from those that worked non-faultly during the second reduced scale test. The Table 2 gives the overview of the the measured damage of the start-up trajectories in reduced-scale model 1) in terms of raw damage, 2) in equivalent time at the Best Efficiency Point (see "Methods"), 3) relative to the Classic trajectory. The results reveal that the optimized trajectory results in a reduction in average damage of over 99.5% compared to classically used trajectories (Classic), and over 70% compared to previously investigated trajectories[16] relying on variable-speed power converters.

The results of damage prediction differ from the ones in Table 1. A few reasons for that are: 1) different sensors working non-faultly during the data acquisition campaigns; 2) in practice the control trajectories do not strictly follow the provided theoretical instructions, e.g., the fluctuations of the real control variables measurements are visible in the Fig. 5a; 3) the optimized trajectory passes through the regions of the phase space without training data available (compare Fig. 4a, b) and model prediction exhibits uncertainty (Fig. 4b).

Qualitatively, the expected improvements were proven to be correct in practice: While running a single Classic trajectory start-up incurs damage equivalent to 111 days, the optimized start-up is only equivalent to 9.5 hours.

## Discussion

We present the results of a data-driven methodology for determining the most effective start-up trajectory that minimizes fatigue damage in hydraulic machines, which is achieved by leveraging a deep learning iTF-DNN model to analyze stress data collected from multiple start-up tests conducted at EPFL-PTMH during a dedicated experimental campaign. Based on iTF-DNN and existing fatigue damage estimation techniques, we employ a pathfinding algorithm to refine the start-up control trajectory based on predicted fatigue levels. The optimized start-up trajectory produced by the optimization framework is defined as a function of the same control parameters typically employed in hydropower stations to steer hydraulic machines. Additionally, the trajectories are designed to comply with the ramping rate constraints of the corresponding full-scale machine, ensuring they can be scaled and implemented in real hydropower plants. We ensure the viability of our optimized control trajectory by conducting another campaign of reduced-scale model tests at EPFL-PTMH. The measurements collected during this campaign confirm a significant reduction in incurred fatigue damage for the optimized control trajectory.

To determine whether we could further improve the optimized trajectory by training the iTF-DNN model on the more extensive data from the second experimental campaign, and re-optimizing the damage, we repeated the modeling process on the second campaign dataset, which included both steady-state and transient data (along with the new trajectory). We re-ran the optimization with the retrained iTF-DNN model to assess any potential improvements in the resulting trajectory. However, the new trajectory showed only slight differences from the one presented in this paper, with a similar damage estimate. We conclude that while further optimization might offer some benefits, these are likely to be relatively minor.

**Table 1 | Damage for optimized start-up based on stress prediction by iTF-DNN vs other start-ups of the first campaign**

| Start-up trajectory | Damage | % of Classic |
|---|---|---|
| Classic | $3.213 \times 10^{-22}$ | 100 |
| Linear | $2.270 \times 10^{-24}$ | 0.71 |
| 2Slopes | $4.989 \times 0^{-24}$ | 1.55 |
| BEP | $1.203 \times 10^{-23}$ | 3.74 |
| optimized | $1.014 \times 10^{-24}$ | 0.32 |

**Table 2 | Damage of start-ups measurements in the second reduced-scale model experimental campaign**

| Start-up trajectory | Damage | Equivalent time at Best Efficiency Point, s | % of Classic |
|---|---|---|---|
| Classic | $(2.555 \pm 1.629) \times 10^{-21}$ | $(9.626 \pm 6.139) \times 10^{6}$ | 100 |
| Linear | $(5.770 \pm 2.726) \times 10^{-22}$ | $(2.174 \pm 1.027) \times 10^{6}$ | 22.6 |
| 2Slopes | $(3.690 \pm 0.610) \times 10^{-23}$ | $(1.390 \pm 0.230) \times 10^{5}$ | 1.44 |
| BEP | $(2.691 \pm 0.348) \times 10^{-23}$ | $(1.014 \pm 0.131) \times 10^{5}$ | 1.05 |
| optimized | $(9.132 \pm 0.462) \times 10^{-24}$ | $(3.441 \pm 0.174) \times 10^{4}$ | 0.36 |

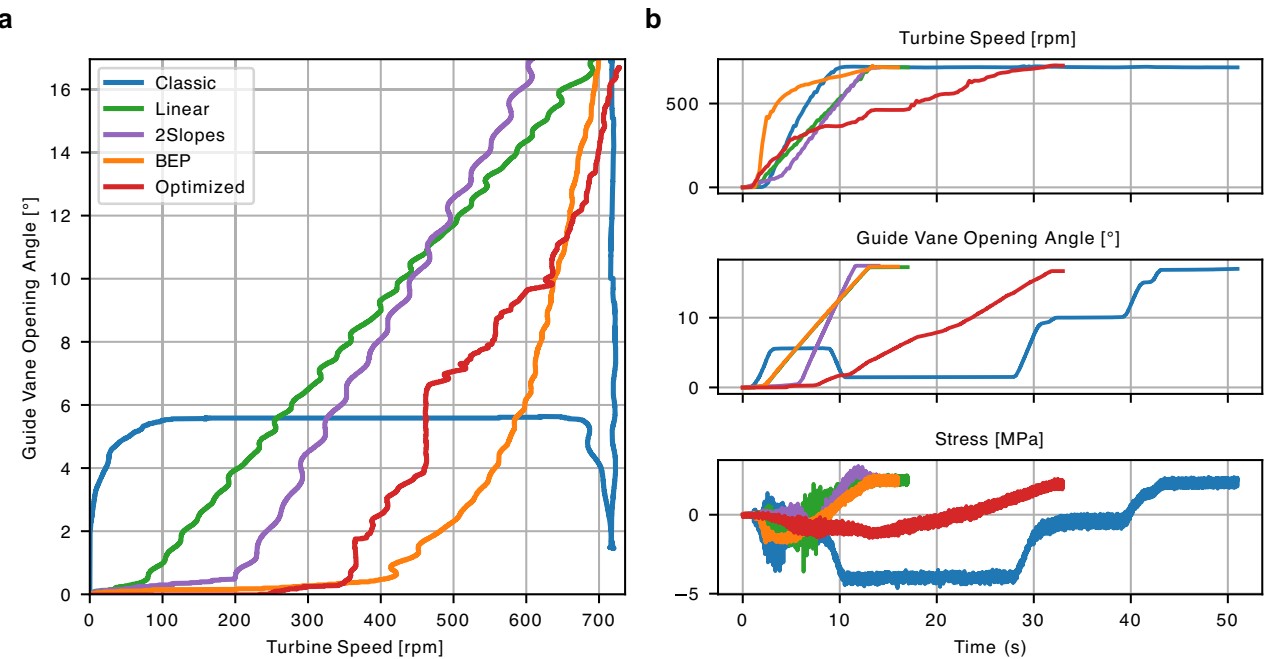

**Fig. 5 | Optimized and existent trajectories collected during the second experimental campaign.** Comparison in the phase space (**a**) and over time (**b**).

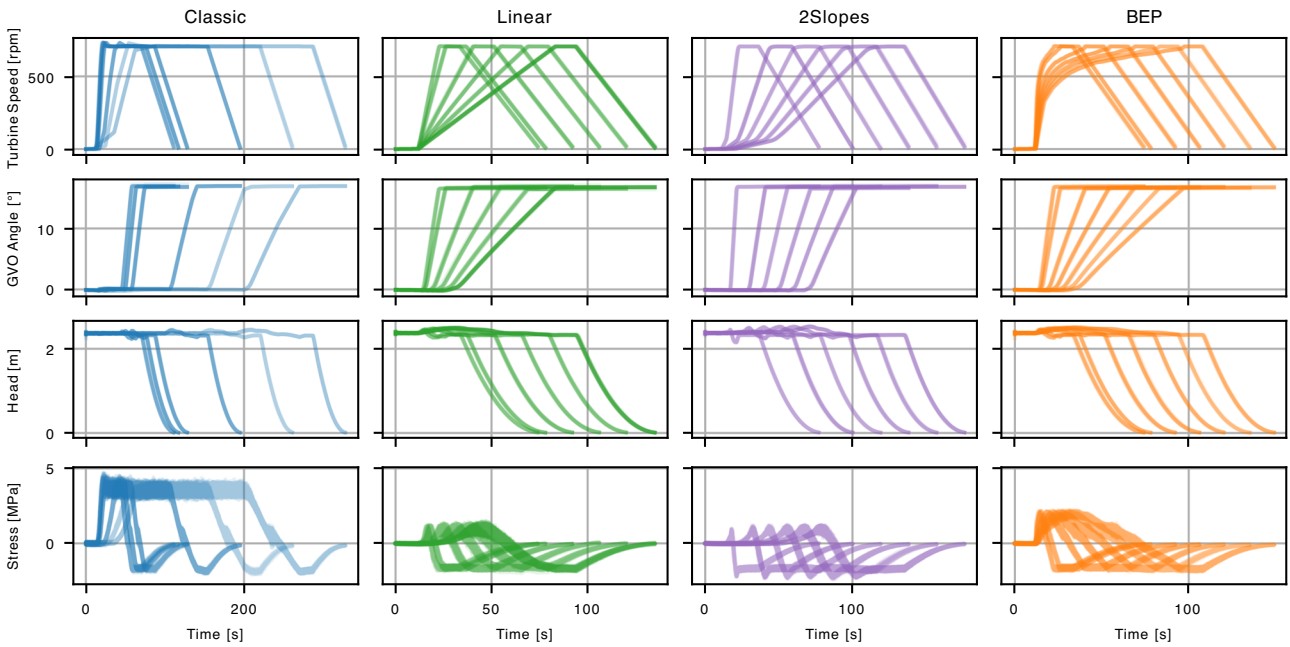

**Fig. 6 | Start-up data of the first experimental campaign.** Summary of the data used for training and validation, plotted by start-up type.

When evaluated in reduced-scale model tests at the EPFL-PTMH platform, the optimized trajectory achieves an average damage reduction of over 99.5% compared to the classically used trajectory, and over 70% compared to previously investigated trajectories that were not optimized to minimize the damage and relied on variable-speed power converters. While the reduced-scale model cannot fully replicate the complete hydraulic response of a hydropower station (in particular the pressure in the hydraulic system and the vibrations at the shaft and casing, due to complexity and interactions among all components of the hydroelectric units), as demonstrated in[19], the stresses on the runner blades obtained from these reduced-scale model tests can be transposed to the full-scale model with fairly good agreement. If stresses were measured on the full-scale machine, as has been done in[40], our proposed methodology would be fully applicable and the stress modeling and trajectory optimization could be recalculated based on an updated database. By minimizing the damage experienced during start-up, the lifespan of critical components can be extended, resulting in improved efficiency and performance of the hydropower system. Our results demonstrate the potential of data-driven methods based on reduced-scale measurements to optimize hydraulic machine operation for sustainable energy generation, paving the way for future studies to transpose and validate the framework in real-world hydropower plants.

## Methods
### Data collection
The first experimental campaign dataset contains measurements of the input and output quantities for about 100 transient operations of a reduced-scale model of Francis turbines for four different startup types. In these tests, strain gauges are attached to both the guide vanes axis and the runner blades to measure the mechanical stresses. Analog signals of the measurements are acquired through a National Instrument (NI) Compact DAQ system and an NI PXI system for pressure fluctuations and stresses. The duration of the time series varied from 7 to 60 seconds with a maximal frequency of measurements of 5000Hz. Some of the observations had different frequencies, which were harmonized during pre-processing by down-sampling all signals to 1000Hz.

The stresses induced in the turbine depend primarily on two control variables: the rotational speed of the runner and the discharge. As the discharge observations suffer from measurement noise, the GVO variable is used, as the characteristic curve of the hydraulic machine bonds the turbine discharge with its guide vane opening passage. In the reduced-scale model platform, a feeding pump attempts to maintain a constant Head during the start-up. In practice, abrupt changes in discharge due to rapid opening of the guide vanes lead to a decrease in Head, which is compensated by a quick increase of the feeding pump speed. Due to the pump's inability to instantly adjust, Head measurements may fluctuate. Hence, to construct the model of the stresses we use the Head, GVO and turbine speed as the inputs. During the trajectory optimization, we vary only two parameters, the Turbine Speed and the GVO, and keep the Head at a fixed level.

The data consists of real measurements of four sensors positioned with the turbine of the reduced scale model, collected for four different start-up types. We examined each type at different time scales, corresponding to Froude numbers in the range of {0.9, 1.0, 1.2, 2.4, 3.6, 4.8, 6, 7.2}. Trajectories shorter in time correspond to smaller Froude numbers, see Fig. 6. Measurements were collected for inputs (GVO, Turbine speed, Head) and outputs (stresses in several positions in the turbine). Each experiment was repeated ten times. The extreme conditions within the turbine sometimes led to sensor failure, causing uncontrolled drifting or high oscillations and rendering some measurements unusable. The model was trained on all runs that did not show visible drifts or other excessive noise. Out of four sensors only two were recorded consistently well, i.e., without persistent drifts and high noise. These sensors were further used in the stress modeling.

The standard Classic start-up sequence corresponds to the trajectory employed with a synchronous generator directly connected to the electrical grid through a transformer: First, the GVO, responsible for the water discharge regulation, is increased up to approximately 20–25% to accelerate the turbine to the synchronous speed; in the second phase, the GVO controller is adjusting the speed of the unit in order to synchronize the generator's phasor with the grid. During this period, the unit is operated under speed-no-load (SNL) conditions, featuring strongly stochastic recirculating flow, low-frequency pressure fluctuations and significant vibrations. Such flows lead to premature wear of the hydraulic machines[41]. Once the synchronization process is completed, the GVO is increased to meet the desired active power production, closing the start-up sequence.

The definitions of the other three available trajectories are tailored to the specific constraints of the power plant case study. First, the maximum GVO and closing rates are restricted by the plant's hydroacoustic parameters for safety measures. Additionally, the unit's rotational speed is set to follow three different acceleration paths:

- The Linear sequence corresponds to a linear increase of both the GVO and the unit's rotational speed to reach the desired operating point. The opening rate of the guide vanes corresponds to the maximum permissible opening rate in order to shorten the sequence as much as possible and to ensure the safe operation of the power plant.
- The BEP trajectory involves adjusting the unit's speed to achieve optimal efficiency as the GVO increases linearly. The speed is regulated throughout the activation sequence based on the turbine's hill chart to maintain this efficiency. Although BEP traditionally refers to the best efficiency point, we use the term BEP trajectory here primarily to describe the trajectory itself, and the meaning should be clear from the context.
- The 2 Slopes sequence is represented by two linear speed increases, respectively two linear GVO increases to mimic the BEP trajectory with simplified unit controller implementation. In the first part of the sequence, the speed is increased progressively to avoid power consumption in accelerating the power group. Only later, at higher speed, a positive turbine's torque is progressively generating power.

After concluding the trajectory optimization, a second campaign of measurements was carried out that included the optimized control trajectory. Across the two reduced-scale data acquisition campaigns, different sets of sensors were successfully recorded, whereas for the input parameters, the observations for both campaigns are of similar quality.

## Damage computation

Damage incurred due to fatigue is estimated following standard mechanical engineering practices: The loading cycles are computed from the stress signal according to the Rainflow-counting method[10]. Following that, the Miner's rule[11] using the Wohler curve[19] quantifies the damage caused by individual load cycles and sums up their contributions to produce a single damage estimate per stress trajectory. The damage computation is carried out for each of the sensors, and the final damage estimate equals the maximum damage for the available sensors.

## iTF-DNN stress model

At each time point for $s$ sensors we observe the vector $\boldsymbol{y}_t \in \mathbb{R}^s$, indexed by time $t = 1, \ldots, T$. For the time series of length $T$ denote stacked vectors $\boldsymbol{y}_t$ as $\boldsymbol{y} \in \mathbb{R}^{s \times T}: \boldsymbol{y} = [\boldsymbol{y}_1, \boldsymbol{y}_2, \ldots, \boldsymbol{y}_T]$. The trend component of the stress is denoted by $\boldsymbol{m} \in \mathbb{R}^{s \times T}$ and the oscillatory high-frequency component is denoted by $\boldsymbol{o} \in \mathbb{R}^{s \times T}$. The control vectors are stored in $\boldsymbol{u} \in \mathbb{R}^{q \times T}$, so that the vector $\boldsymbol{u}_t \in \mathbb{R}^q$ represents the control values at time $t$. The value of $q$ equals 3 during model training when all inputs are used, and $q = 2$ during optimization, as one of the variables (the Head) is kept fixed. For multiple trajectories of observed measurements, we add additional indexing for the variables, e.g., $y_{j,t}, t = 1, \ldots, T_j$, where $j$ runs through the trajectory number. The stress depends on the input parameters, and we omit this dependence in the notation.

To model stresses and optimize damage, we assume that stress trends depend instantaneously on control parameters. This means that at any given time point, the values of GVO, turbine speed, and Head are sufficient to fully determine the average stress. With this in mind, we propose the following decomposition for the observed stress time series.

$$\boldsymbol{y}_t = \boldsymbol{m}_t + \boldsymbol{o}_t + \boldsymbol{\epsilon}_t, \quad t = 1, \ldots, T, \tag{3}$$

where $\boldsymbol{m}_t$ is a slowly varying trend, $\boldsymbol{o}_t$ is an oscillatory component, and $\boldsymbol{\epsilon}_t$ is random homoscedastic noise. The stress depends on $\boldsymbol{u}_t \in \mathbb{R}^q$, $q = 3$, as additionally to GVO and turbine speed, the Head level was added due to fluctuations of the measurements from the constant value.

Our modeling choice fell on neural networks due to their ability to capture complex, nonlinear relationships in highly oscillating time series that depend on control variables. At the same time learning the high-frequency oscillations is challenging[42]: A basic neural network model tends to learn low frequencies primarily and can struggle with approximations of high frequencies. Our work is inspired by the PFF-DNN[43] and PhaseDNN[44] models to improve the learning of high-frequency signals. Both models attempt to learn the signal representation in the time-frequency domain by mimicking the Short-Time Fourier Transform (STFT) of the signal (or the residual after modeling the signal trend). Note that these models aim to learn an approximation of the observed signal using neural networks with no dependence on input variables. Our case is more complex, as we are given time-varying input parameters that affect the signal dynamics and, in particular, its STFT. In what follows, we propose a model that aims to learn the dependence of the oscillatory signal dynamics on the available input control parameters. We abbreviate our model as input Time-Frequency deep neural network (iTF-DNN).

We assume $\boldsymbol{m}_t$ is deterministic and depends only on the current controls $\boldsymbol{u}_t$, and does not depend on the previous control history. The remaining oscillatory component $\boldsymbol{o}_t$ is allowed to depend on control history in a non-instantaneous way. In particular, we estimate the amplitude and phase of the STFT of the remaining signal (after having estimated $\boldsymbol{m}_t$) depending on the control values in the neighboring STFT windows. The estimate of the oscillatory component $\boldsymbol{o}_t$ is further recovered through an ISTFT based on the models of amplitudes and phases of the STFT. We describe the models more formally below.

For damage estimation in the case of a large spread of $\boldsymbol{m}_t$ values, the damage induced by $\boldsymbol{o}_t$ is much smaller, as according to Miner's rule, low-frequency oscillations with large amplitudes typically have a more significant impact on cumulative fatigue damage compared to high-frequency oscillations with smaller amplitudes. Nevertheless, to obtain a general model of stresses and estimate damage precisely, it is important to model both the trend $\boldsymbol{m}_t$ and the oscillations $\boldsymbol{o}_t$ accurately.

**Trend model.** We model $\boldsymbol{m}_t$ by a neural network (ResNet[20]) $f_{\boldsymbol{m}, \boldsymbol{\theta}_m}(\boldsymbol{u}_t)$, where $\boldsymbol{u}_t$ are the input control parameters and $\boldsymbol{\theta}_m$ are the parameters of the neural network, which are optimized by the minimization of the mean squared error (MSE) between targets $\boldsymbol{y}_t$ and the network's predictions $NN_m(\boldsymbol{u}_t)$ on the training set of $p$ time series of lengths $T_k$, $k = 1, \ldots, p$ until convergence:

$$\hat{\boldsymbol{\theta}}_0 = \arg \min_{\boldsymbol{\theta}} \sum_{k=1}^{p} \sum_{t=1}^{T_k} (y_{k,t} - f_{\boldsymbol{m}, \boldsymbol{\theta}}(u_{k,t}))^2 \tag{4}$$

to obtain a preliminary estimate of the trend $f_{\boldsymbol{m}, \hat{\boldsymbol{\theta}}_0}(\boldsymbol{u})$.

**Oscillatory component.** Next, we estimate the residual

$$\hat{\boldsymbol{o}}_t = \boldsymbol{y}_t - f_{\boldsymbol{m}, \hat{\boldsymbol{\theta}}_0}(\boldsymbol{u}_t) \tag{5}$$

between the observed stress value and the trend estimates to obtain a de-trended oscillatory signal. We model the amplitude and phase of the oscillatory residual by approximating its STFT with neural network models dependent on the controls. For simplicity of notation, assume $s = 1$, the extension to larger dimensions $s$ of the output is straightforward. Consider the STFT of the residual signal $\hat{\boldsymbol{o}}$ for one trajectory: Given a window size $W$ and hop size $S$, the STFT components $\mathcal{F}(\hat{\boldsymbol{o}})[i, l] \, l = 1, \ldots, L, L = \lfloor (T - W)/S \rfloor$, $i = 1, \ldots, \lfloor W/2 \rfloor$ are calculated for $L$ windows of length $W$ with the hop $S$ and a set of frequencies, indexed by $i$. We fixed the length of one window to $W = 1024$ and $S = 256$. For each STFT window, we get vectors of

**Table 3 | Performance of the iTF-DNN**

| Start-up trajectory | $R^2$ | MSE | Bias | (Min, Max) |
|---|---|---|---|---|
| **Train** | | | | |
| Classic | 0.995 ± 0.001 | 0.025± 0.007 | 0.007 ± 0.054 | (−1.972, 4.971) |
| Linear | 0.985 ± 0.006 | 0.010 ± 0.004 | −0.032 ± 0.033 | (−1.877, 1.516) |
| 2Slopes | 0.992 ± 0.003 | 0.006 ±0.001 | −0.020 ± 0.020 | (−2.381, 1.322) |
| **Validation** | | | | |
| BEP | 0.976 ± 0.008 | 0.028 ±0.015 | −0.082 ± 0.041 | (−1.897, 2.433) |

amplitudes $\boldsymbol{a}_l$ and phases $\boldsymbol{\phi}_l$ with components

$$a_{l,i} = |\mathcal{F}(\hat{\boldsymbol{o}})[i,l]|, \quad \phi_{l,i} = \arctan\left(\frac{\text{Im}(\mathcal{F}(\hat{\boldsymbol{o}})[i,l])}{\text{Re}(\mathcal{F}(\hat{\boldsymbol{o}})[i,l])}\right), \quad i = 1, \ldots, \lfloor W/2 \rfloor. \tag{6}$$

To model the amplitude and phase vectors $\boldsymbol{a}_l$ and $\boldsymbol{\phi}_l$, we train two vector-output ResNet neural networks $f_{\boldsymbol{a},\boldsymbol{\theta}_a}$ and $f_{\boldsymbol{\phi},\boldsymbol{\theta}_\phi}$. These networks take a control history $\bar{\boldsymbol{u}}_l$, corresponding to the $l$-th STFT window, as input and are trained by minimizing the error in the frequency domain, combined with with the trend modeling error, using the initialization $\hat{\boldsymbol{\theta}}_0$ for the trend model parameters:

$$\hat{\boldsymbol{\theta}}_m, \hat{\boldsymbol{\theta}}_a, \hat{\boldsymbol{\theta}}_\phi = \arg\min_{\boldsymbol{\theta},\boldsymbol{\theta}_a,\boldsymbol{\theta}_\phi} \sum_{k=1}^{p} \sum_{t=1}^{T_k} (y_{k,t} - f_{m,\boldsymbol{\theta}}(u_{k,t}))^2 \tag{7}$$

$$+ \alpha \sum_{i=1}^{\lfloor W/2 \rfloor} \sum_{k=1}^{p} \sum_{l=1}^{L_k} (a_{i,l,k} - f_{\boldsymbol{a},\boldsymbol{\theta}_a}(\bar{\boldsymbol{u}}_{k,l}))^2 \tag{8}$$

$$+ \lambda \sum_{i=1}^{\lfloor W/2 \rfloor} \sum_{k=1}^{p} \sum_{l=1}^{L_k} \sin\left[\frac{1}{2}(\phi_{i,l,k} - f_{\boldsymbol{\phi},\boldsymbol{\theta}_\phi}(\bar{\boldsymbol{u}}_{k,l}))^2\right], \tag{9}$$

where $\alpha$ and $\lambda$ are hyperparameters.

To obtain the estimates of $\boldsymbol{o}$ from the STFT estimate $\hat{\mathcal{F}}_{\hat{\boldsymbol{\theta}}_o}(\boldsymbol{u})$, $\hat{\boldsymbol{o}}_o = \{\hat{\boldsymbol{\theta}}_a, \hat{\boldsymbol{\theta}}_\phi\}$, given by the models of the amplitude $f_{\boldsymbol{a},\hat{\boldsymbol{\theta}}_a}(\bar{\boldsymbol{u}}_l)$ and phase $f_{\boldsymbol{\phi},\hat{\boldsymbol{\theta}}_\phi}(\bar{\boldsymbol{u}}_l)$, $l = 1, \ldots, L$, we apply an ISTFT to get $f_{\boldsymbol{o},\hat{\boldsymbol{\theta}}_o}(\boldsymbol{u}, t) = \mathcal{F}^{-1}[\hat{\mathcal{F}}_{\hat{\boldsymbol{\theta}}_o}(\boldsymbol{u})]_t$, such that $f_{\boldsymbol{o},\hat{\boldsymbol{\theta}}_o} : \mathbb{R}^{\lfloor W/2 \rfloor \times L} \to \mathbb{R}^T$. After each optimization gradient step, the residual is estimated by (5) for the updated parameters of the trend model.

The final iTF-DNN model combines the model of the trend and the model of the oscillation

$$f_{NN}(\boldsymbol{u}_t) = f_{m,\hat{\boldsymbol{\theta}}_m}(\boldsymbol{u}_t) + f_{\boldsymbol{o},\hat{\boldsymbol{\theta}}_o}(\boldsymbol{u}, t). \tag{10}$$

During inference, we perform forward passes through the networks to obtain trend and oscillatory predictions for a given control trajectory.

**Model performance.** The iTF-DNN model performance results on training and validation on the data from the first experimental campaign are shown in Table 3, namely mean and standard deviation of per-trajectory $R^2$, MSE, Bias (average of non-absolute residual values) of predictions together with (Min, Max) of observations for reference. The model has a tendency to have a small negative bias. Overall, training and validation results show good agreement between the errors on training and validation datasets.

## Dijkstra optimization

Using the fitted iTF-DNN model, we implement an optimization procedure that systematically explores the space of possible trajectories to find the lowest possible damage trajectory. For the damage optimization process, we focus on the sensor exhibiting the best data quality.

Consider stresses and input control trajectories in the interval of time $[0, T]$, sampled with the time step $\Delta\tau$ (at 1kHz, $\Delta\tau = 1$ms). Note that the Head level was used to train the iTF-DNN, as it can fluctuate in the reduced-scale model, but during the trajectory optimization, we keep the Head fixed and optimize the damage in the Turbine Speed and the GVO. Thus, the stress depends only on two inputs: Turbine speed and GVO, that is $\boldsymbol{u}_t \in \mathbb{R}^q$, where $q = 2$.

Recall that we are trying to solve (2), the problem of finding the minimally damaging sequence of controls $\boldsymbol{u}^\star \in \mathcal{U}$, where $\mathcal{U}$ is a space of viable trajectories, i.e., each sequence $\boldsymbol{u}^\circ \in \mathcal{U}$ fulfills the constraints:

1. Boundary Conditions: The sequence starts at a standstill and ends at operating conditions ($\boldsymbol{u}_{\text{op}}$),

$$\boldsymbol{u}_0^\circ = 0, \quad \boldsymbol{u}_T^\circ = \boldsymbol{u}_{\text{op}}, \tag{11}$$

where $T$ is not limited to a specific trajectory length.

2. Bounded increments: Neither the Turbine speed $n$ nor the GVO $g$ decrease at any point and the trajectory does not exceed machine limits on turbine acceleration and GVO velocity;

$$0 \le \boldsymbol{u}_{t+1}^\circ - \boldsymbol{u}_t^\circ \le \Delta\boldsymbol{u}. \tag{12}$$

3. Allowed: The trajectory lies completely within an allowed phase space $\mathcal{C}$,

$$\boldsymbol{u}_t^\circ \in \mathcal{C} \subset \mathbb{R}^2. \tag{13}$$

This problem is low-dimensional, continuous, non-convex, and bounded.

Dijkstra's Algorithm[22] is a widely used algorithm for solving the shortest path problem, i.e., the problem of finding the minimal-cost path between two nodes in a weighted graph. Starting from an initial node $n_0$, the algorithm iteratively considers the closest visited node $\tilde{n}$, and updates the distances to the nodes $n_i(\tilde{n})$ connected to this node. The algorithm terminates when the entire graph has been visited, or when a target node has been reached. By visiting the nodes in order of increasing cost, the algorithm is guaranteed to produce an optimal solution. Consider a discrete problem where the turbine speed and GVO take values on a regular grid, resulting in a collection of evenly-spaced nodes. To ensure the computational feasibility of the trajectory search, we consider coarsened trajectories, by effectively reducing the frequency of the data points by $K$. Thus, we introduce a timescale by

defining the increment $\Delta t = K\Delta\tau$. To construct the graph for the Dijkstra optimization, we set all the nodes of the grid to be its vertices: For a 2D grid defined by equidistant points along the intervals $[0, u_{op,1}]$ and $[0, u_{op,2}]$, with $N$ equidistant points along each axis, the total number of points on the grid is $N^2$. Each point $(x_{1,i}, x_{2,j})$ on the grid can be represented as:

$$x_{1,i} = i\frac{u_{op,1}}{N}, \quad x_{2,j} = j\frac{u_{op,2}}{N}, \quad \text{for } i,j = 0, \ldots, N. \quad (14)$$

For each vertex, we draw edges to all nodes where a transition would not violate the bounded increments condition (12): Let $n_{i,j}$ be the node at the position $(x_{1,i}, x_{2,j})$. An edge exists between two nodes $n_{i,j}$ and $n_{k,l}$ if and only if the following coordinate-wise inequality is satisfied:

$$0 < x_{1,i} - x_{1,k} \leq K\Delta u_1 \quad \text{and} \quad 0 < x_{2,j} - x_{2,l} \leq K\Delta u_2. \quad (15)$$

Next, we compute the weights for the edges: We set the edge weights to correspond to the increase in damage incurred by extending the path from a node $n^*$ to a neighbor $\bar{n}$. By setting the cost to the increase in damage, we achieve the desirable property that the cost to access $\bar{n}$ is exactly the damage of the least-damaging trajectory that ends at $\bar{n}$. To reconstruct the trajectory in the original time scale $\Delta\tau$, for the calculation of the weight, we fill in the control values between two nodes and linearly interpolate between them at a rate of $1/\Delta\tau$.

### Algorithm 1. Dijkstra-based Trajectory Optimization

**Require:** start_node, goal_node, priority_queue, cost, parent, visited
1: **While** priority_queue is not empty **do**
2: (current_cost, current_node) ← heappop(priority_queue)
3: **if** current_node == goal_node **then**
4: break
5: **end if**
6: **for** neighbor in get_neighbors(current_node) **do**
7: path ← reconstruct_path(parent, current_node) + neighbor
8: trajectory ← interpolate_trajectory(path)
9: stress ← calculate_stress(trajectory)
10: new_cost ← compute_damage(stress)
11: **if** neighbor not in cost **or** new_cost < cost[neighbor] **then**
12: cost[neighbor] ← new_cost
13: parent[neighbor] ← current_node
14: heappush(priority_queue, (new_cost, neighbor))
15: **end if**
16: **end for**
17: **end while**
18: optimal_path ← reconstruct_path(parent, goal_node)
19: **return** optimal_path, cost[goal_node]

The described graph is used for the trajectory optimization problem (2) with the help of Dijkstra's Algorithm 1 with the starting node set to (0, 0) and the end and limit nodes $\boldsymbol{u}_{op} = (736, 17)$. Note that the maximum start-up duration, $T_{max}$, is related to $N$ and $K$ by $2N \cdot (K \text{ms}) = T_{max}$ and that the number of edges that need to be explored per node is of the order $E_{node} \propto K^2N^2$ due to (15). The total computational complexity of the algorithm is dominated by the number of edges $E_{node}N^{2\,45}$. Combining these observations, the complexity scales as $E_{node}N^2 \propto N^2$. We have selected the time resolution $K = 128$ and the number of points $N = 256$, by manual adjustment as a compromise between the computational complexity and the time discretisation step of the optimized trajectory, given a maximum start-up duration $T_{max} = 65s$.

To ensure viable trajectories are produced, we impose a limit on turbine acceleration and GVO angular velocity (12), corresponding to

machine limits, which are 280rpms⁻¹ and 2.16°s⁻¹ correspondingly. Additionally, we disallow parts of the phase space (13) since they would require active braking of the turbine to traverse or contain regions where the trend iTF-DNN is particularly uncertain about its prediction. See Fig. 4a for the disallowed region.

### Equivalent time at Best Efficiency point quantification

At best efficiency point (BEP) the turbine converts the specific hydraulic energy and the angular momentum of the flow into mechanical energy with minimal losses, thus reaching the BEP. This depends on the hydraulic machine geometry and control parameters of the operating condition. The equivalent time at BEP expresses the operational time of a hydraulic turbine as if it had been running continuously at its ideal conditions (BEP)[16]. To quantify the equivalent time at BEP, we rely on steady-state stress measurement collected during the second campaign. By analyzing a 20-second stress measurement collected at the BEP, we calculate the damage incurred per second. We divide the damage incurred during one start-up by this value to derive the equivalent time at BEP. Note that in this study the term BEP trajectory is reserved for a start-up sequence, tracking the BEP for each rotational speed during the machine acceleration from 0 to nominal rotational speed.

## Data availability

The raw data from the stress measurements are protected and are not available due to data privacy laws. The optimized control trajectory data generated in this study and the trained model are available on GitHub[46].

## Code availability

The modeling and optimization code, as well as a demo using a simplified artificial dataset are available on GitHub[46].

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

## Acknowledgements

The research leading to the results published in this paper was supported by PAIRED HYDRO, Project C21-11, granted within the 5th SDSC Call for Collaborative Data Science Projects. We thank Guillaume Obozinski for enlightening discussions.

## Author contributions

E.V. conceived the original research idea and sourced the data. M.S. acquired the data. E.V. and A.M. provided insights about the data and assumptions. E.K. supervised the machine learning research and, together with T.M., designed the machine learning approach. T.M. built and validated the model and optimization scheme, producing the visuals of its results. T.M. and E.K. wrote the initial manuscript, T.M., E.K., A.M., E.V. edited it.

## Funding

## Competing interests

The authors declare no competing interests.
