## [Transparent Peer Review file · Nature Communications]

Fatigue Damage Reduction in Hydropower Startups with Machine Learning

Corresponding Author: Dr Ekaterina Krymova

Version 0:

Reviewer comments:

Reviewer #1

(Remarks to the Author)

This paper discusses a method of Fatigue Damage Reduction for hydropower systems during the transient start-up. The research topic is meaningful from the perspective of engineering operation and maintenance. However, the design idea of this paper is still a little crude for solving practical problems in hydropower startups.

First, the fatigue damage of hydraulic machines in the transient start-up is due to the unit passing through the vibration zone (i.e., a small load interval). For example, A 300 MW hydropower system may have a vibration zone between 0 and 130MW. The vibration zone is the operation deviating from the design condition, which is the occurrence and development areas of the blade vortex. Vibrations in the shaft, swing deviations in guide bearings, and pipeline pressure pulsation are the key factors that cause fatigue damage in hydraulic machines. However, the method presented in this paper is realized by only optimizing two input trajectories, including the turbine speed and guide vane opening. The changes in these two input trajectories cannot fully measure the fatigue damage of hydraulic machines, and some necessary trajectories from Dynamic Balance Test like the vibration of turbine head cover, swing of guide-vane bearing, and water pressure at volute inlet should be considered as inputs. In addition to this, the author can also consider combining shafting vibration theory with data mining to solve the problem of fatigue damage, which may improve reliability in engineering applications.

Second, the optimized inputs are curves with real-time fluctuation characteristics in Fig. 1, which is difficult to adopt in practical hydropower operations. Due to the characteristic of time delay in water flow and hydraulic machinery, further demonstrations are needed to realize the coordinated starting of guide vane and turbine speed.

Based on the above comments, a major revision is needed for the current version of the paper.

(Remarks on code availability)

Reviewer #2

(Remarks to the Author)

The work presents an interesting approach for minimizing damage during the startup of hydroelectric generating units by optimizing their startup trajectories (from the unit being completely stopped to the conditions of maximum efficiency), using a data-driven model for damage estimation. I consider the work suitable for acceptance pending minor revisions.

Key results

The results show a significant reduction compared to the classic startup strategy and relevant improvements over other strategies.

Validity

The interpretation of the data and conclusions appear to have been conducted in a valid and robust manner.

Significance

The work provides a significant contribution to the field, introducing a methodology that can mitigate the impacts caused by the increased number of startups due to changes in electrical systems with the integration of intermittent energy sources into the grid.

Data and methodology

The methodology and the quality of the presentation are appropriate, except for the points discussed below.

Analytical approach

The approach used seems appropriate given the innovative nature of the work. It may be beneficial to include performance metrics obtained from the model during training (R^2 , MSE, or RMSE).

Clarity and context

Overall, the text is clear and well-contextualized. However, there are some points that could improve the clarity of the work:

- The abbreviation GVO first appears on line 133 but is only explained on line 284 (and appears several times within this range). I believe the abbreviation should not appear before being explained, and the explanation should only be given once on line 133. In subsequent mentions, only the abbreviation should be used.
- In Table 2, in the column title "Time at best Efficiency Point," it seems that "Equivalent" is missing, which affects the immediate understanding.
- Figure 3 is never referenced in the text. If the information it contains is important, the text should reflect this; if not, the figure is irrelevant.

Suggested improvements

I believe the text would benefit from a brief explanation of what the startup damages are and how they relate to the observed variables (GVO and turbine speed) during the contextualization.

It would be beneficial to discuss how the deterioration of the turbine and generator components is expected to affect the model's stress estimation.

An explanation on the rationale for the choice of $k = 128$ on line 458 would be beneficial.

References

The manuscript appropriately references previous literature.

(Remarks on code availability)

The provided repository includes a README file with instructions and provides an apparently functional environment. However, I was unable to reproduce the model because the dataset was not available as per the README instructions. The code contains basic comments and appears to be sound for reproduction, analysis, and understanding. Nonetheless, I could not verify its functionality due to the issue mentioned. Another consideration is the time required for reproduction; the authors note that training and execution on a NVIDIA P100 GPU environment could take few days to be completed. Since I do not have access to such resources, the environments I can access have worse performance, thus the time required would be even greater.

Reviewer #3

(Remarks to the Author)

(Remarks on code availability)

All my comments were compiled together with my co-reviewer's and sent through his review report.

Version 1:

Reviewer comments:

Reviewer #1

(Remarks to the Author)

The comments were handled by authors.

(Remarks on code availability)

Reviewer #2

(Remarks to the Author)

The concerns raised in the first round of review have been thoroughly addressed, resulting in significant improvements to the manuscript. The inclusion of detailed performance metrics, such as R^2 , MSE, and Bias, for training and validation datasets enhances the credibility of the work. The clarity of the text has been improved by appropriately introducing the abbreviation "GVO" and ensuring its consistent use, as well as by clarifying the title of Table 2 and properly referencing Figure 3. Additionally, the expanded introduction provides a clear explanation of startup damage and its relationship with the control variables, while the discussion on the impact of turbine and generator component deterioration on stress estimation adds depth to the analysis. The comprehensive justification for the choice of the value k demonstrates a careful consideration of computational complexity and time resolution.

(Remarks on code availability)

Reviewer #3

(Remarks to the Author)

(Remarks on code availability)

NCOMMS-24-66399-T: Response to the Referees

December 10, 2024

Dear Reviewers,

Thank you very much for the time spent on your feedback and thoroughly checking the manuscript. Your feedback was very valuable and has certainly helped improve our submission. We would like to thank all the three reviewer for the comments on the paper. We have made an effort to try and address the main points they raised. In the following sections, we provide detailed point by point responses to the comments.

Reviewer 1

1. *This paper discusses a method of Fatigue Damage Reduction for hydropower systems during the transient start-up. The research topic is meaningful from the perspective of engineering operation and maintenance. However, the design idea of this paper is still a little crude for solving practical problems in hydropower startups.*

First, the fatigue damage of hydraulic machines in the transient start-up is due to the unit passing through the vibration zone (i.e., a small load interval). For example, A 300 MW hydropower system may have a vibration zone between 0 and 130MW. The vibration zone is the operation deviating from the design condition, which is the occurrence and development areas of the blade vortex. Vibrations in the shaft, swing deviations in guide bearings, and pipeline pressure pulsation are the key factors that cause fatigue damage in hydraulic machines. However, the method presented in this paper is realized by only optimizing two input trajectories, including the turbine speed and guide vane opening. The changes in these two input trajectories cannot fully measure the fatigue damage of hydraulic machines, and some necessary trajectories from Dynamic Balance Test like the vibration of turbine head cover, swing of guide-vane bearing, and water pressure at volute inlet should be considered as inputs. In addition to this, the author can also consider combining shafting vibration theory with data mining to solve the problem of fatigue damage, which may improve reliability in engineering applications.

Thank you for your valuable feedback. The optimization in our study is based on a database containing strain measurements on the runner blades, taken from a fully homologous (both hydraulically and mechanically) reduced-scale model, in compliance with the IEC standard. This model captures the stresses arising from fluid-structure interactions and vibrations within the system, and these recorded stresses are incorporated into the optimization process as they determine the search space. We chose the opening angle of

the guide vanes and the rotational speed as the control variables, as these are the only adjustable parameters during the operation of a Francis-type hydraulic machine. Parameters, such as the vibration of the turbine head cover, swing of the guide-vane bearing, and water pressure at the volute inlet, cannot be directly controlled and, therefore, cannot be considered as control variables. However, a database including the measurement of the stresses directly on the full scale machine would include the effect of all these parameters on the runner fatigue-induced damage and our model and optimization framework would still be fully applicable (likely with an optimized trajectory slightly different to the one that we found with our database).

We recognize that the full hydraulic response of a hydropower station cannot be fully replicated in a reduced-scale model. However, as demonstrated in the PhD thesis of Dr. Seydoux [2], the stresses obtained from these reduced-scale model tests can be transposed to the full scale with a fairly good agreement. We have clarified our hypothesis throughout the paper including a discussion on the origin of the stresses (lines 49, 55-67), the rationale behind the choice of control parameters (lines 61-65, 140-143), the limitations associated with the hydraulic response of the piping system and possible practical implementations enabling future works (lines 102-104, 264-272). This should provide greater clarity regarding the assumptions and limitations of our model.

2. *Second, the optimized inputs are curves with real-time fluctuation characteristics in Fig. 1, which is difficult to adopt in practical hydropower operations. Due to the characteristic of time delay in water flow and hydraulic machinery, further demonstrations are needed to realize the coordinated starting of guide vane and turbine speed.*

Thank you for the comment. The optimization trajectories presented in Fig. 1 have been implemented and tested on the reduced-scale model platform. These trajectories were designed to respect the ramping rate constraints of the homologous full-scale machine using a full-size frequency converter, ensuring practical feasibility as these trajectories are implemented in the PID control system of the hydroelectric unit. The trajectories were implemented by considering the real-time fluctuations of the reduced scale model platform and this did not compromise the minimization of fatigue-induced damage. On a full-scale application, the trajectory must be scaled in time, and, therefore, the dynamics of the transient will be slower than the one tested on the reduced scale model. We have included a discussion in the paper to clarify this aspect and to address the practicality of implementing the optimized trajectories in real-world hydropower operations (lines 102-104, 242-247, 264-272).

Reviewer 2

1. *The approach used seems appropriate given the innovative nature of the work. It may be beneficial to include performance metrics obtained from the model during training (R^2 , MSE, or RMSE).*

Thank you for the comment, in the following table you could find the results of the model performance on training and validation. The model has a tendency towards a small negative bias. Overall, training and validation results show good agreement between the

	Start-up trajectory	R^2	MSE	Bias	(Min, Max)
Train	Classic	0.995 ± 0.001	0.025 ± 0.007	0.007 ± 0.054	(-1.972, 4.971)
	Linear	0.985 ± 0.006	0.010 ± 0.004	-0.032 ± 0.033	(-1.877, 1.516)
	2Slopes	0.992 ± 0.003	0.006 ± 0.001	-0.020 ± 0.020	(-2.381, 1.322)
Validation	BEP	0.976 ± 0.008	0.028 ± 0.015	-0.082 ± 0.041	(-1.897, 2.433)

Table 1: Mean and standard deviation of per-trajectory R^2 , MSE, Bias (average of non-absolute residual values) of predictions, and (Min, Max) of observations.

errors on training and validation datasets. We included this information in the lines 165-167, 435-438.

2. **Clarity and context** Overall, the text is clear and well-contextualized. However, there are some points that could improve the clarity of the work:

The abbreviation GVO first appears on line 133 but is only explained on line 284 (and appears several times within this range). I believe the abbreviation should not appear before being explained, and the explanation should only be given once on line 133. In subsequent mentions, only the abbreviation should be used.

We thank the reviewer for their remark. We have changed the text following their suggestion, defining the term when first introduced in line 64 and using the abbreviation thereafter. We retain the non-abbreviated form whenever not referring to the control variable, (e.g. in the context of discussing the guide vane opening rate limitations).

3. In Table 2, in the column title "Time at best Efficiency Point," it seems that "Equivalent" is missing, which affects the immediate understanding.

Thank you for the comment, we added "Equivalent" to the column name of the Table 2.

4. Figure 3 is never referenced in the text. If the information it contains is important, the text should reflect this; if not, the figure is irrelevant.

We thank the reviewer for their attention to detail. This was caused by an erroneous duplicate label for figures. We have fixed this problem (line 170), and Figure 3 is now referenced correctly in the text.

5. **Suggested improvements** I believe the text would benefit from a brief explanation of what the startup damages are and how they relate to the observed variables (GVO and turbine speed) during the contextualization.

It would be beneficial to discuss how the deterioration of the turbine and generator components is expected to affect the model's stress estimation.

Thank you for the suggestions. In the introduction, line 49 and 55-67, we have added an explanation in the paper to clarify the nature of startup damages and their relationship

to the controlled variables, specifically the guide vane opening (GVO) and turbine speed. This addition aims to provide better understanding of how these variables influence the stresses and potential fatigue damage in the turbine.

6. *An explanation on the rationale for the choice of $K = 128$ on line 458 would be beneficial.*

To ensure the computational feasibility of the trajectory search, we consider coarsened trajectories, by effectively reducing the frequency of the data points by K . We have selected the time resolution K and number of points N , defining the phase space grid by manual adjustment as a compromise between the computation complexity and time-discretisation step of the optimized trajectory for the given maximum start-up duration. Let us consider the computational complexity of the optimization, which is approximately proportional to the number of edges, as in our case the number of edges is larger than the number of nodes and dominates the complexity [1]. Consider the nodes in the $N \times N$ grid with the steps $\Delta x = \frac{u_{\text{lim},1}}{N}$ and $\Delta y = \frac{u_{\text{lim},2}}{N}$ for x - and y -axis correspondingly. From the condition on the number of edges (9) in the paper, $0 < x_i - x_k \leq K\Delta u_1$ and $0 < y_j - y_l \leq K\Delta u_2$ the number of edges to explore for each node is of the order $K^2\Delta u_1\Delta u_2/(\Delta x\Delta y) \propto K^2N^2$. Given that there are N^2 nodes, the total number of edges, and thus computational complexity is roughly $\propto N^4K^2$.

Note, that due to the monotonous increase condition (7) for the maximal start-up trajectory T_{max} , it holds that $2N \cdot (K\text{ms}) = T_{\text{max}}$, where for convenience we omit the units further. Thus, for a fixed T_{max} , the final computational complexity of the optimization is $\propto N^2$, i.e. depends solely on N .

We considered the powers of 2 for N and K . A choice of $K = 128$ fixes the interpolation time between two nodes to 128ms and for a 256×256 grid it gives a maximum runtime of 65s: $(256 + 256) * (0.128\text{s}) \approx 65\text{s}$. This runtime aligns well with the intended time-scale for the slowest possible outcome of the algorithm.

When using 256×256 grid and $K = 128$, the algorithm had a runtime of several days. Considering that the optimization only needs to be ran once per machine configuration, we found this duration acceptable. In contrast, running the algorithm on a more fine-grained grid would likely take much longer to complete with our computational set-up. For example, a 512×512 grid with $K = 64$ could be another option delivering the same T_{max} . At the same time from the consideration above, if we aim to keep the maximal time fixed and increase the time resolution twice (decrease K twice), the corresponding compute increases 4 times.

We have added the details in the lines 469-472, 490-498.

7. *(Remarks on code availability):*

The provided repository includes a README file with instructions and provides an apparently functional environment. However, I was unable to reproduce the model because the dataset was not available as per the README instructions. The code contains basic comments and appears to be sound for reproduction, analysis, and understanding. Nonetheless, I could not verify its functionality due to the issue mentioned. Another consideration is the time required for reproduction; the authors note that training and execution on a NVIDIA

P100 GPU environment could take few days to be completed. Since I do not have access to such resources, the environments I can access have worse performance, thus the time required would be even greater.

We are grateful to the reviewer for taking the time to check the code. We have added a jupyter notebook which showcases the end-to-end process, from training a model to running the optimization using this model and mentioned this in the lines 522-523. To facilitate running this code on a CPU-only machine, we supply a small set of training data and configured the optimization of the trajectory to run on a coarsened search grid. Note, that we are not allowed to make the real data available. For this reason, to verify reproducibility, the provided data examples used for training the model consist of real control trajectories but artificially created stress trajectories.

References

- [1] Michael L Fredman and Robert Endre Tarjan. *Fibonacci heaps and their uses in improved network optimization algorithms*. Journal of the ACM (JACM), 34(3):596–615, 1987.
- [2] Martin Seydoux. *Study of flexible operating conditions in variable-speed hydraulic turbines: advanced models and experimental validation*. Technical report, EPFL, 2024.

NCOMMS-24-66399-T: Response to the Referees

January 15, 2025

Dear Reviewers,

Thank you for your time and feedback. We believe that it has helped us to significantly improve the presentation of our results.